

# A model based on Rock-Eval thermal analysis to quantify the size of the centennially persistent organic carbon pool in temperate soils

Lauric Cécillon[1,2], François Baudin[3], Claire Chenu[4], Sabine Houot[4], Romain Jolivet[2], Thomas Kätterer[5], Suzanne Lutfalla[2], Andy Macdonald[6], Folkert van Oort[4], Alain F. Plante[7], Florence Savignac[3], Laure Soucémarianadin[2], Pierre Barré[2]

[1]Université Grenoble Alpes, Irstea, UR EMGR, F-38402 St-Martin-d'Hères, France
[2]Laboratoire de Géologie de l'ENS, PSL Research University, CNRS UMR 8538, F-75005, Paris, France
[3]Sorbonne Université-UPMC-Univ Paris 06, Institut des Sciences de la Terre de Paris, F-75005 Paris, France
[4]AgroParisTech - INRA, UMR 1402 ECOSYS, F-78850 Thiverval-Grignon, France
[5]Department of Ecology, Swedish University of Agricultural Sciences, 750 07 Uppsala, Sweden
[6]Department of Sustainable Agriculture Sciences, Rothamsted Research, Harpenden, Hertfordshire, AL5 2JQ, UK
[7]Earth and Environmental Science, University of Pennsylvania, Philadelphia, USA

*Correspondence to*: Lauric Cécillon (lauric.cecillon@irstea.fr)

**Abstract.** Changes in global soil carbon stocks have considerable potential to influence the course of future climate change. However, a portion of soil organic carbon (SOC) has a very long residence time (> 100 years) and may not contribute significantly to terrestrial greenhouse gas emissions during the next century. The size of this persistent SOC reservoir is presumed to be large. Consequently, it is a key parameter required for the initialization of SOC dynamics in ecosystem and Earth system models, but there is considerable uncertainty in the methods used to quantify it. Thermal analysis methods provide cost-effective information on SOC thermal stability that has been shown to be qualitatively related to SOC biogeochemical stability. The objective of this work was to build the first quantitative thermal analysis based model of the size of the centennially persistent SOC pool. We used a unique set of soil samples from four agronomic experiments in Northwestern Europe with long-term bare fallow and non-bare fallow treatments (*e.g.* manure amendment, cropland and grassland), as a sample set for which estimating the size of the centennially persistent SOC pool is relatively straightforward. At each experimental site, we estimated the average concentration of centennially persistent SOC and its uncertainty by applying a Bayesian curve fitting method on the observed declining SOC concentration over the duration of the long-term bare fallow treatment. Overall, the estimated concentrations of centennially persistent SOC ranged from 5 to 11 gC.kg-1 soil (lowest and highest boundaries of four 95% confidence intervals). Then, by dividing site-specific concentrations of persistent SOC by the total SOC concentration of 118 archived soil samples from long-term bare fallow and non-bare fallow treatments, we could estimate the proportion of centennially persistent SOC in the samples and the associated uncertainty. The proportion of centennially persistent SOC ranged from 0.14 (standard deviation of 0.01) to 1 (standard deviation of 0.15). Samples were subjected to thermal analysis by Rock-Eval 6 that generated a series of 30 parameters reflecting their SOC thermal stability and bulk chemistry. The sample set was split into a calibration set (n = 88) and a validation set (n = 30). We trained a non-parametric machine learning algorithm (random forests multivariate regression model) that accurately





predicted the size of the centennially persistent SOC pool using Rock-Eval 6 thermal parameters as predictors in the calibration set (pseudo-$R^2$ = 0.91, RMSEC = 0.06) and the validation set ($R^2$ = 0.91, RMSEP = 0.07). The uncertainty of the predictions obtained using the multivariate regression model was quantified by a Monte Carlo approach that produced conservative 95% prediction intervals across the 30 samples of the validation set. This model based on Rock-Eval 6 thermal

analysis can thus be used to predict the proportion of centennially persistent SOC with known uncertainty in new soil samples from similar pedoclimates. Our study strengthens the evidence for a link between the thermal and biogeochemical stability of soil organic matter, and demonstrates that Rock-Eval 6 thermal analysis can be used to quantify the size of the centennially persistent organic carbon pool in temperate soils.

## 1 Introduction

Soils exert a key regulation of the atmospheric greenhouse gas concentrations on a decadal timescale through the net carbon source and sink status of their organic carbon reservoir (Amundson, 2001; Eglin et al., 2010). However, a portion of the soil organic carbon (SOC) reservoir may not contribute significantly to the net exchange of $CO_2$ and $CH_4$ between atmosphere and land during the next century because its residence time exceeds 100 years and its rate of carbon inputs is low (Trumbore, 1997; He et al., 2016). The size of this centennially persistent SOC pool is presumed to be large (*i.e.* between one and two

thirds of total SOC) and dependent on geochemical parameters such as soil texture and mineralogy (Buyanovsky and Wagner, 1998a; Trumbore, 2009; Mills et al., 2014; Mathieu et al., 2015). However, the amount of centennially persistent organic carbon in soils is highly uncertain as it cannot be estimated accurately by current analytical methods (Post and Kwon, 2000; von Lützow et al., 2007; Bruun et al., 2008). Physico-chemical procedures attempting to isolate SOC with high residence time from bulk SOC have proven unsatisfactory because of indications that such fractions are a mixture of ancient

and recent SOC (von Lützow et al., 2007; Trumbore, 2009; Lutfalla et al., 2014). Even the well-established radiocarbon ($^{14}$C) analytical technique cannot precisely determine the size of the centennially persistent SOC pool (Schrumpf and Kaiser, 2015; Menichetti et al., 2016). The importance of better information on the size of the centennially persistent SOC pool has been emphasized recently (International Soil Carbon Initiative, 2011; Bailey et al., in press; Bispo et al., 2017; Harden et al., in press), stressing the need for operational and standardized metrics or proxies to accurately quantify SOC persistent at the

centennial timescale. The general lack of information on the size and turnover rate of measurable SOC pools hampers the initialization of SOC pools in dynamic models, questioning their predictions of the evolution of the global SOC reservoir (Falloon and Smith, 2000; Luo et al., 2014; Feng et al., 2016; He et al., 2016; Sanderman et al., 2016). Luo et al. (2016) and He et al. (2016) recently claimed that optimizing parameter estimation with global data sets on SOC pools and fluxes was the highest priority to reduce biases among Earth system models.

During the past decade, thermal stability of organic carbon has been proposed as a good surrogate for its biogeochemical stability in litter and soils (*e.g.* Rovira et al., 2008; Plante et al., 2009; Gregorich et al., 2015). Several studies using thermal analysis techniques such as thermogravimetry and differential scanning calorimetry during ramped combustion have shown



that the fast-cycling SOC pool determined as the amount $CO_2$ respired in laboratory incubation experiments was thermally labile (Plante et al., 2011; Leifeld and von Lützow, 2014; Campo and Merino, 2016). Recently, studies using thermal analysis under oxidative or inert (pyrolysis) reaction atmosphere coupled with evolved gas analysis have shown a high and positive correlation between the thermally stable SOC and persistent SOC determined using [14]C measurements (Plante et al.,

2013), and between thermally stable SOC and mineral-associated SOC isolated by a classical SOC physical fractionation scheme (Saenger et al., 2015). Using long-term bare fallow (LTBF) soils kept free of vegetation for several decades (*i.e.* with negligible carbon inputs), Barré et al. (2016) recently showed that persistent SOC was low in energy and thermally stable. While there appears to be strong qualitative links between thermal and biogeochemical stability of SOC, there is to date no established quantitative link between the size of the persistent SOC pool and SOC thermal characteristics.

The objective of this work was to design a reliable, routine method based on a thermal analysis technique (Rock-Eval 6; RE6) to quantify centennially persistent SOC in a range of soils. First, we compiled a set of reference soil samples from four long-term agronomic experiments in Northwestern Europe with long-term bare fallow treatments. The SOC concentration of LTBF treatments can be used to estimate the size of the persistent SOC pool of a particular site, as proposed by Rühlmann (1999) and Barré et al. (2010). Here, we refined estimates of the persistent SOC concentration previously published by Barré

et al. (2010) for the four sites used in this study. We then used these values to estimate the proportion of centennially persistent SOC in 118 archived soil samples from LTBF and non-LTBF treatments of these four sites. The last step consisted in analyzing these reference samples using RE6 thermal analysis, and building a multivariate regression model to relate RE6 information on SOC thermal stability and bulk chemistry to the estimated proportion of centennially persistent SOC. In this work, we aimed at delivering a thermal analysis based model with reliable prediction intervals around the predicted values of

the size of the centennially persistent SOC pool. We thus had a particular focus on the uncertainty of the estimated proportion of centennially persistent SOC and its propagation in the multivariate regression model.

## 2 Materials and methods

### 2.1 Reference soil sample set with estimated size of the centennially persistent SOC pool

The reference soil sample set was built using samples from four long-term agronomic experimental sites in Northwestern

Europe (Versailles, France, Grignon, France, Rothamsted, United Kingdom, Ultuna, Sweden, Supplementary material S1). Each of the four sites includes a LTBF treatment, with bare fallow durations ranging from 48 years at Grignon to 79 years at Versailles. For all experimental sites, we also included non-LTBF treatments that have increased or maintained their total SOC concentrations over time, or sustained smaller losses than the LTBF treatment. The selected non-LTBF treatments included manure amendments (Versailles), straw or composted straw amendments (Grignon), continuous grassland

(Rothamsted), and continuous cropland (Ultuna). Soil samples from each site and treatment have been regularly collected and archived since the initiation of the experiments. A total of 118 topsoil (0–20 to 0–25 cm depth, Supplementary material S1) samples were selected from the archives of LTBF and non-LTBF treatments to build the reference sample set. Samples




were selected from two or three field replicate plots in each decade from the initiation of the experiments up to 2007 (Grignon), 2008 (Versailles, Rothamsted) or 2009 (Ultuna) to obtain a sample set with the widest possible range of proportions of centennially persistent SOC. The non-LTBF treatments and multiple sites also added to the diversity of land-use, climate and parent material. For each sample, total SOC concentration was measured by dry combustion with an

5 elemental analyzer ($SOC_{EA}$, gC.kg$^{-1}$ soil) after decarbonatation when necessary according to NF ISO 10694 (1995).

Based on the decline in total SOC concentration over the duration of the LTBF treatment, Barré et al. (2010) estimated the concentration of centennially persistent SOC at each site using a Bayesian curve fitting method applied to each LTBF field replicate plot. Here, we refined those site-specific estimates by (i) applying a similar Bayesian curve fitting method on combined SOC concentration data from all LTBF field replicate plots of each site (four field replicate plots for Ultuna and

10 Rothamsted, six field replicate plots for Versailles and Grignon), and (ii) using new SOC concentration data up to 2014 for Rothamsted and 2015 for Ultuna, increasing their LTBF duration to 55 years for Rothamsted and 59 years for Ultuna.

For each site, we assumed the temporal evolution of LTBF SOC concentration, γ(t), followed an exponential decay function:

$$\gamma(t) = ae^{-bt} + c \,, \tag{1}$$

where γ(t) (unit = gC.kg$^{-1}$ soil) is the LTBF SOC concentration at time t, t (unit = year) is the time under bare fallow, and a, b and c are fitting parameters. Parameter a (unit = gC.kg$^{-1}$ soil) corresponds to the amplitude of the decay and b (unit = y$^{-1}$) is the characteristic decay rate. The parameter c (unit = gC.kg$^{-1}$ soil) represents a theoretically inert portion of SOC. We considered this parameter as a site-specific metric of the centennially persistent SOC concentration. We used a Bayesian inference method to compute site-specific estimates of centennially persistent SOC concentration and associated

uncertainties (detailed in the Section 2.3.1).

The proportion of centennially persistent SOC ($CP_{SOC}$) in each soil sample was then calculated as the ratio of the site-specific $CP_{SOC}$ concentration to the total SOC concentration of the sample:

$$CP_{SOC} \text{ proportion [sample]} = \frac{CP_{SOC} \text{ concentration [site]}}{SOC \text{ concentration [sample]}} \,, \tag{2}$$

where the unit of $CP_{SOC}$ concentration [site] and SOC concentration [sample] is gC.kg$^{-1}$ soil. The $CP_{SOC}$ proportions of five samples that were slightly above 1 were set to 1. In these calculations, we assumed that at each site, the concentration of $CP_{SOC}$ was the same in the LTBF and non-LTBF treatments and was constant with time. The details related to the estimation of the uncertainty on the $CP_{SOC}$ proportion of each sample are reported in the Section 2.3.2.

## 2.2 Thermal analysis of soil samples by Rock-Eval 6

The 118 soil samples from the reference set were analyzed with a RE6 Turbo device (Vinci Technologies) using the basic set-up for the analysis of soil organic matter (Behar et al., 2001; Disnar et al., 2003). The RE6 technique provided measurements from the sequential pyrolysis and oxidation of *ca.* 40 mg of finely ground (< 250 µm) soil per sample (Fig. 1).





Volatile hydrocarbon effluents from pyrolysis were detected and quantified with flame ionization detection (FID), while the evolution of CO and $CO_2$ gases were quantified by infrared detection during both pyrolysis and oxidation stages. Pyrolysis was carried out from 200 °C to 650 °C in a $N_2$ atmosphere with a heating rate of 30 °C minute$^{-1}$, while the oxidation was carried out from 300 °C to 850 °C in a laboratory air atmosphere (with $O_2$) with a heating rate of 20 °C minute$^{-1}$. The RE6

technique generated five thermograms per sample (Fig. 1, *i.e.* volatile hydrocarbon (HC) effluents during pyrolysis, $CO_2$ during pyrolysis, $CO_2$ during oxidation, CO during pyrolysis, and CO during oxidation). On average, the organic carbon yield of the RE6 analysis was greater than 96.5% of $SOC_{EA}$ for the soils of the reference sample set ($SOC_{RE6} = 0.966 \times SOC_{EA} + 0.403$, $R^2 = 0.97$, n = 118).

For each RE6 thermogram, we determined the temperatures corresponding to each incremental proportion of the amount of

10 gases evolved during the pyrolysis and oxidation stages. Upper temperatures of 850 °C (CO oxidation thermogram), 650 °C (HC pyrolysis thermogram), 611 °C ($CO_2$ oxidation thermogram) and 560 °C (CO and $CO_2$ pyrolysis thermograms) were chosen for signal integration (Fig. 1), thereby excluding any interference of soil carbonates (Behar et al., 2001). Thermal decomposition of carbonates was indeed observed beyond 560 °C (CO and $CO_2$ pyrolysis thermograms) and 611 °C ($CO_2$ oxidation thermogram) for the site of Grignon (data not shown). For the three pyrolysis thermograms, signal integration

started after an isotherm step of 200 s at 200 °C. Finally, we retained 5 of these temperature parameters per thermogram: $T_{10}$, $T_{30}$, $T_{50}$, $T_{70}$, $T_{90}$ which respectively represent the temperatures corresponding to the evolution of 10, 30, 50, 70, and 90% of the amount of evolved gases for each sample and for each of the five different thermograms (HC, $CO_2$ pyrolysis, $CO_2$ oxidation, CO pyrolysis, CO oxidation).

For the HC pyrolysis thermogram we also determined three parameters reflecting a proportion of thermally resistant or labile

hydrocarbons: a parameter representing the proportion of hydrocarbons evolved between 200 °C and 450 °C (thermo-labile hydrocarbons, TLHC-index, modified from Saenger et al., 2015), the I-index representing the preservation of thermally labile immature hydrocarbons (after Sebag et al., 2016), and the R-index representing the proportion of hydrocarbons thermally stable at 400°C (after Sebag et al., 2016). Those three RE6 parameters were calculated as follows:

$$\text{TLHC-index} = \frac{\text{Area of HC pyrolysis thermogram [200 °C–450 °C]}}{\text{Total area of HC pyrolysis thermogram}}, \tag{3}$$

$$\text{I-index} = \log_{10}\left(\frac{\text{proportion of HC pyrolysis thermogram [200 °C–400 °C]}}{\text{proportion of HC pyrolysis thermogram [400 °C–460 °C]}}\right), \tag{4}$$

$$\text{R-index} = \frac{\text{Area of HC pyrolysis thermogram [400 °C–650 °C]}}{\text{Total area of HC pyrolysis thermogram}}, \tag{5}$$

Using the HC pyrolysis thermogram, we determined a parameter reflecting SOC bulk chemistry, the hydrogen index (HI, mgHC.g$^{-1}$C), that corresponds to the quantity of pyrolyzed hydrocarbons relative to $SOC_{RE6}$. Using the CO and $CO_2$





pyrolysis thermograms, we determined another parameter reflecting SOC bulk chemistry, the oxygen index (OI$_{RE6}$, mgO$_2$.g$^{-1}$C) corresponding to the oxygen yield as CO and CO$_2$ during thermal pyrolysis of soil organic matter divided by the total SOC (SOC$_{RE6}$) of the sample. The HI correlates with the elemental H:C atomic ratio of SOC and the OI$_{RE6}$ correlates with the elemental O:C atomic ratio of SOC (Espitalié et al., 1977).

Overall, we thus calculated for each soil sample a series of 30 RE6 parameters reflecting SOC thermal stability and bulk chemistry to be used in subsequent statistical and modelling analyses.

Signal integration of the RE6 thermograms and calculation of the RE6 temperature parameters were performed with R v.3.4.3 (R Core Team, 2017) and the hyperSpec (Beleites and Sergo, 2014), pracma (Borchers, 2015) and stringr (Wickham, 2015) packages.

**2.3 Statistical analysis**

**2.3.1 Bayesian inference of site-specific CP$_{SOC}$ concentrations and uncertainties**

At each site, the CP$_{SOC}$ concentration was estimated as the model parameter c of the exponential decay function described in Eq. (1). To estimate the value of this parameter and assess its uncertainty, we sampled the posterior Probability Density Function (PDF) of the model parameters in Eq. (1), which is given by Bayes' theorem as a function of the prior PDF (*i.e.* what we know before collecting data) and the likelihood (*i.e.* how likely is it to predict the data given a set of parameters).

The posterior PDF is the combination of our prior knowledge and of the information carried by the data, including measurement uncertainties. For a model vector **m** (containing the parameters a, b and c) and a data vector **d** of all the measurements of SOC concentrations, the posterior PDF, P(**d** | **m**), is P(**d** | **m**) $\propto$ P(**m**)P(**m** | **d**), with P(**m**) the prior PDF on the model parameters and P(**m** | **d**) the likelihood.

We chose uniform PDFs for the model parameters, a, b, and c to be as uninformative as possible. We use the Gaussian form of the likelihood function, such as P(**m** | **d**) $\propto e^{-\frac{1}{2}\left(\mathbf{d} - \gamma(\mathbf{t})\right)^{\mathbf{T}}\mathbf{C_d}^{-1}\left(\mathbf{d} - \gamma(\mathbf{t})\right)}$, where **t** is the vector of all observation times and **C**$_d$ is the data covariance matrix describing the uncertainties on the SOC measurements. We consider a conservative standard deviation for SOC concentration data (0.75 gC.kg$^{-1}$ soil) estimated by Barré et al. (2010) for the same soils. We use a Metropolis algorithm to draw $3 \times 10^4$ samples from the posterior PDF with a burning phase of $3.7 \times 10^5$ steps. We can then

derive the mean and standard deviation for the parameter c from the posterior PDF.

**2.3.2 Estimating the uncertainty of CP$_{SOC}$ proportion in each sample**

Based on our assessment of the uncertainties on SOC concentration data and site-specific CP$_{SOC}$ concentrations (see above), we propagated these errors to estimate the uncertainty on the CP$_{SOC}$ proportion in each soil sample. This was estimated by calculating the standard deviation of the CP$_{SOC}$ proportion for each sample as follows:

sd (CP$_{SOC}$ proportion [sample]) =



$$\text{CPsoc proportion [sample]} \times \sqrt{\left(\frac{\text{sd(CPsoc concentration [site])}}{\text{CPsoc concentration [site]}}\right)^2 + \left(\frac{\text{sd(SOC concentration [sample])}}{\text{SOC concentration [sample]}}\right)^2},\qquad(6)$$

where abbreviation sd stands for standard deviation.

### 2.3.3 Statistical relationships between RE6 parameters and $CP_{SOC}$ proportion

The reference sample set was randomly split into a calibration set (n = 88 samples) and a validation set (n = 30 samples). The
correlations between the 30 RE6 parameters and the $CP_{SOC}$ proportion were assessed with a non-parametric Spearman's rank
correlation test on the calibration set (n = 88). A principal component analysis (PCA) of the 30 centered and scaled RE6
parameters was performed for the calibration set to (i) summarize the variance of SOC thermal stability and bulk chemistry
on a single factorial map, and (ii) illustrate the correlations among RE6 parameters. Correlations between the $CP_{SOC}$
proportion in calibration soils and their principal component scores were determined using Spearman's rank correlation tests,
and its relationships with the 30 RE6 parameters were further illustrated by projecting the $CP_{SOC}$ proportion variable in the
PCA correlation plot. The RE6 data of the soils from the validation set were projected on the same PCA factorial map to
check that the validation set was representative of the calibration set.

### 2.3.4 Random forests regression model to predict $CP_{SOC}$ proportion from RE6 parameters

A multivariate regression model was built to relate $CP_{SOC}$ proportion in the reference samples from the calibration set
(response vector or dependent variable $\mathbf{y}$, n = 88) to their SOC thermal stability and bulk chemistry, summarized by a matrix
of predictor variables ($\mathbf{X}$) made of the 30 centered and scaled RE6 parameters. The non-parametric and non-linear machine
learning technique of random forests (RF, Breiman, 2001; Strobl et al., 2009) was used to build this model. The random
forests regression model was based on a forest of 1000 diverse regression trees made of splits and nodes. A random forests
learning algorithm combines bootstrap resampling and random variable selection. Each of the 1000 regression trees was thus
grown on a bootstrapped subset of the calibration set (*i.e.* containing about two thirds of "in-bag" calibration samples) by
randomly sampling 10 out of the 30 RE6 parameters as candidates at each split of the tree, and using a minimum size of
terminal tree nodes of five soil samples. The random forests regression model was then used to predict the proportion of
$CP_{SOC}$ in the validation set (n = 30), a prediction corresponding to the mean of the predicted values across the 1000
regression trees.

The performance of the random forests regression model for predicting $CP_{SOC}$ proportion was assessed by statistics
comparing the RF-predicted *vs.* reference (estimated) values of the sample set. The performance statistics were calculated
on: (i) the "out-of-bag" soil samples of the calibration set and (ii) the soil samples of the validation set. Out-of-bag samples
are observations from the calibration set not included in the learning sample set for a specific tree that can be used as a
"built-in" test set for calculating its prediction accuracy (Strobl et al., 2009). The performance statistics included the
coefficient of determination (pseudo-$R^2$ for the calibration set or $R^2$ for the validation set) and the root-mean-square error of
calibration or prediction (RMSEC for the calibration set or RMSEP for the validation set). The ratio of performance to



deviation (RPD) and the bias of the random forests regression model were additionally calculated for the validation set. The relative importance (*i.e.* ranking) of each of the 30 RE6 parameters for the prediction of the proportion of $CP_{SOC}$ in the RF regression model was computed as the unscaled permutation accuracy (Strobl et al., 2009).

### 2.3.5 Error propagation in the random forests regression model

Since our objective was to deliver a thermal analysis based model with reliable prediction intervals around the predicted values of the $CP_{SOC}$ proportion, we estimated the prediction uncertainty of the random forests model for new soil samples. We used a methodology recently published by Coulston et al. (2016) to approximate prediction uncertainty for random forests regression models, and adapted it to explicitly take into account the uncertainty on reference values of $CP_{SOC}$ proportion (Eq. (6)) that were used to build the model (Supplementary material S2).

Briefly, we sampled with replacement (*i.e.* bootstrapped) the calibration set (**y, X**) 2000 times to obtain 2000 bootstrap samples ($\mathbf{y}^{*b}$, $\mathbf{X}^{*b}$) that were used to parametrize 2000 random forest models ($\mathbf{RF}^{*b}$). To incorporate the uncertainty on reference values of $CP_{SOC}$ proportion, each of the 2000 bootstrapped vectors ($\mathbf{y}^{*b}$) contained values of $CP_{SOC}$ proportion that were randomly sampled from normal distributions with means and standard deviations of the $CP_{SOC}$ proportion of the corresponding soil samples from the calibration set (Eq. (6)). For each bootstrap sample of the calibration set, resampling

discarded approximatively 37% of the data ($\mathbf{y}^{*-b}$, $\mathbf{X}^{*-b}$) that were used for prediction. We obtained an error assessment dataset made of 2000 vectors of observed (reference) values $\mathbf{y}^{*-b}$, predicted values $\bar{\hat{y}}^{*-b}$ (mean of the predictions across 1000 regression trees for each observation), and $\mathbf{var(\hat{y})}^{*-b}$ (variance of the predictions across 1000 regression trees for each observation). For each observation of the 2000 bootstrap samples, we calculated a metric $\tau$ allowing to scale between $\mathbf{var(\hat{y})}$ that can be calculated for any soil sample by the random forests regression model, and the squared prediction error $(\mathbf{y} - \bar{\hat{y}})^2$

that is only available for the reference sample set. The metric $\tau$ was calculated as follows (Coulston et al., 2016):

$$\tau = \sqrt{\frac{(y - \bar{\hat{y}})^2}{var(\hat{y})}},\tag{7}$$

A Monte Carlo approach was used to estimate $\hat{\tau}$, the 95[th] percentile of all calculated $\tau$ values for all out-of-bag observations of the 2000 bootstraps (Supplementary material S2). This $\hat{\tau}$ value was such that 95% of the predictions of the $CP_{SOC}$ proportion lie within $\hat{\tau} \times \mathbf{sd(\hat{y})}$ of the true value of $CP_{SOC}$ proportion (*i.e.* 95% prediction intervals). As $\mathbf{sd(\hat{y})}$, the standard

deviation of the predictions of the $CP_{SOC}$ proportion across 1000 regression trees, can be calculated by the random forests regression model for any soil sample, this approach allows the calculation of 95% prediction intervals on any new soil sample for which only X data (30 RE6 parameters) are available. We calculated the 95% prediction intervals ($\bar{\hat{y}} \pm \hat{\tau} \times \mathrm{sd}(\hat{y})$) for the validation set (n = 30) to examine whether those intervals included the true (estimated) values of $CP_{SOC}$ proportion. More details on the procedure to approximate prediction uncertainty for random forests regression models are provided in

Coulston et al. (2016). We finally checked how the error on $CP_{SOC}$ proportion propagated into the random forests regression model by (i) comparing the value of $\hat{\tau}$ with or without incorporating the uncertainty on reference values of $CP_{SOC}$ proportion in the algorithm, and (ii) by comparing the sizes of the 95% prediction intervals calculated for the validation soil samples



with their respective 95% confidence intervals (determined by multiplying their standard deviation calculated in Eq. (6) by 1.96).

The Bayesian inference method was performed with Python 2.7 and the PyMC library (Patil et al., 2010). All other statistical analyses were performed with R v.3.4.3 (R Core Team, 2017) and the factoextra package for running PCA (Kassambara, 2015), the randomForest package for running the random forests regression models (Liaw and Wiener, 2002) and the boot package for bootstrapping (Davison and Hinkley, 1997; Canty and Ripley, 2015).

# 3 Results

## 3.1 $CP_{SOC}$ concentration at each site and $CP_{SOC}$ proportion in reference soil samples

The Bayesian inference of the parameter c of the exponential decay function (Eq. (1)) yielded site-specific estimates of the $CP_{SOC}$ concentration with 95% confidence intervals (Eq. (1), Table 1, Fig. 2). Estimated $CP_{SOC}$ concentrations ranged from 6.22 gC.kg$^{-1}$ soil at Versailles to 10.46 gC.kg$^{-1}$ soil at Rothamsted. The uncertainty on $CP_{SOC}$ concentration was lower at Rothamsted (standard deviation of 0.27 gC.kg$^{-1}$ soil) and Versailles (standard deviation of 0.31 gC.kg$^{-1}$ soil) than at Ultuna (standard deviation of 0.88 gC.kg$^{-1}$ soil) and Grignon (standard deviation of 1.00 gC.kg$^{-1}$ soil).

Overall, the wide range in total SOC concentrations within and across sites (from 5 to 46 gC.kg$^{-1}$ soil, Table 1) combined with an assumed constant $CP_{SOC}$ concentration within each site, resulted in a reference sample set with a wide spectrum of $CP_{SOC}$ proportions ranging from 0.14 to 1 (Eq. (2), Table 1). The uncertainty (standard deviation) on the values of $CP_{SOC}$ proportion ranged from 0.01 to 0.15 for the reference sample set (Eq. (6), Supplementary material S3). High uncertainties were found for high values of $CP_{SOC}$ proportion (*i.e.* samples with longer time periods under bare fallow treatment), with a modulation by the site-specific $CP_{SOC}$ concentration uncertainty (Grignon > Ultuna > Versailles > Rothamsted, Table 1), as expected from Eq. (6) (Supplementary material S3).

The random splitting of the reference sample set generated calibration and validation sample sets with similar mean values, range of values and standard deviations for both total SOC concentration and $CP_{SOC}$ proportion (Table 1).

## 3.2 Relationships between RE6 parameters and $CP_{SOC}$ proportion

The 30 RE6 parameters showed contrasted correlations with the $CP_{SOC}$ proportion in the calibration set (Table 2). Most RE6 temperature parameters showed positive correlations with the $CP_{SOC}$ proportion, with Spearman's ρ above 0.8 for four of them (the RE6 temperature parameter corresponding to 50% of $CO_2$ gas evolution during the pyrolysis stage, $T_{50\_CO2\_PYR}$, and the RE6 temperature parameters corresponding to 30%, 50% and 70% of $CO_2$ gas evolution during the oxidation stage, $T_{30\_CO2\_OX}$, $T_{50\_CO2\_OX}$, $T_{70\_CO2\_OX}$, Table 2).

Conversely, five RE6 temperature parameters showed significant negative correlations with the $CP_{SOC}$ proportion ($T_{10\_HC\_PYR}$, $T_{10\_CO\_PYR}$, $T_{30\_CO\_PYR}$, $T_{50\_CO\_PYR}$, $T_{70\_CO\_PYR}$, Table 2). Out of the three RE6 parameters reflecting a proportion of thermally resistant or labile hydrocarbons, only the TLHC-index showed a weakly significant negative Spearman's ρ with



the $CP_{SOC}$ proportion, the I-index and the R-index showing no correlations (Table 2). The two RE6 parameters reflecting SOC bulk chemistry showed highly significant correlations with the $CP_{SOC}$ proportion (Table 2), the HI being negatively correlated and the $OI_{RE6}$ being positively correlated.

The PCA of the centered and scaled RE6 parameters illustrates the correlations among those 30 variables in the calibration
set (Fig. 3). A continuum of $CP_{SOC}$ proportion values was observed in the reference samples along the first two principal components (Fig. 3A), and projecting the $CP_{SOC}$ proportion in the PCA correlation circle further highlighted the relationships between this variable and the 30 RE6 parameters (Fig. 3B). The $CP_{SOC}$ proportion variable had a strongly negative projected loading score on PC1 (Fig. 3B), as well as negative projected loadings on PC2 (Fig. 3B) and PC3 (data not shown). The scores of the calibration soils on the first three principal components were indeed significantly and negatively correlated with
the $CP_{SOC}$ proportion ($\rho$ = -0.71, $p$-value < 0.001 for PC1, $\rho$ = -0.36, $p$-value < 0.001 for PC2, $\rho$ = -0.25, $p$-value < 0.05 for PC3), such that a large part (82%) of the variance of the 30 RE6 parameters was linked to the $CP_{SOC}$ proportion in the calibration set.

The random splitting of the reference sample set generated calibration and validation sample sets with similar RE6 thermal characteristics as illustrated by their similar distribution on the factorial map of the first two principal components of the
15 PCA (Fig. 3A).

### 3.3 Performance of the regression model using RE6 parameters to predict $CP_{SOC}$ proportion

The random forests regression model performed very well in predicting the $CP_{SOC}$ proportion in the reference sample set using the 30 RE6 parameters as predictors (Fig. 4). Both performance statistics on the calibration set (pseudo-$R^2$ = 0.91, RMSEC = 0.06, n = 88) and on the validation set ($R^2$ = 0.91, RMSEP = 0.07, n = 30) demonstrated the good predictive
power of the regression model based on RE6 thermal analysis.

Propagating the estimated uncertainties on the values of $CP_{SOC}$ proportion increased the size of the prediction intervals of RE6-RF regression model. Indeed, the value of $\hat{\tau}$ increased from 1.72 to 2.10 when the uncertainty on $CP_{SOC}$ proportion was integrated in the algorithm described at Section 2.3.5. The horizontal and vertical error bars on Fig. 4 illustrate the global error propagation on the $CP_{SOC}$ proportion estimates in the RE6-RF regression model for the validation soil sample set. The
25 values of the total width of the 95% confidence interval (reference estimations of $CP_{SOC}$ proportion, horizontal error bars in Fig. 4) were 0.03 (minimum total width), 0.58 (maximum total width) and 0.24 (mean total width) for the soil samples of the validation set (n = 30). For the 95% prediction intervals (RE6-RF predictions of $CP_{SOC}$ proportion, vertical error bars in Fig. 4), the uncertainties increased to 0.14 (minimum total width), 0.67 (maximum total width) and 0.35 (mean total width). The thirty 95% prediction intervals for RE6-RF predictions of $CP_{SOC}$ proportion in the validation set all included their respective
reference estimation of $CP_{SOC}$ proportion (Fig. 4).

Out of the 30 RE6 parameters tested by the random forests model as possible predictor variables of the $CP_{SOC}$ proportion in the calibration set, the RE6 temperature parameters corresponding to 50% of $CO_2$ gas evolution during the pyrolysis stage ($T_{50\_CO2\_PYR}$) and to 30% of $CO_2$ gas evolution during the oxidation stage ($T_{30\_CO2\_OX}$) showed the highest importance scores





(based on permutation accuracy importance calculations, Table 2). The twelve most important RE6 parameters for predicting the $CP_{SOC}$ proportion were temperature parameters calculated on the five different RE6 thermograms (Table 2). The two RE6 parameters reflecting SOC bulk chemistry ($OI_{RE6}$ and HI) were of medium importance to predict the $CP_{SOC}$ proportion, while the RE6 parameters reflecting a proportion of thermally resistant or labile hydrocarbons (I-index, R-index and TLHC-index)
were of weak importance (Table 2).

## 4 Discussion

### 4.1 A unique soil sample set with accurate and contrasted values of $CP_{SOC}$

Adding new SOC concentration data for Rothamsted (up to 2014) and Ultuna (up to 2015), and combining SOC concentration data from all LTBF field replicate plots of each site decreased the uncertainty on the site-specific estimates of
the $CP_{SOC}$ concentration (Fig. 2), compared with the previous estimations published by Barré et al. (2010). Indeed, the total width of the 95% confidence interval around the estimation of the site-specific $CP_{SOC}$ concentration slightly decreased from 1.4 to 1.2 gC.kg$^{-1}$ soil at Versailles and from 4.96 to 3.92 gC.kg$^{-1}$ soil at Grignon, and strongly decreased from 7.24 to 3.46 gC.kg$^{-1}$ soil at Ultuna and from 5.98 to 1.06 gC.kg$^{-1}$ soil at Rothamsted (Table 1, Fig. 2, Barré et al., 2010). The mean estimated values of the $CP_{SOC}$ concentration were marginally changed at Versailles (6.22 *vs*. 6.12 gC.kg$^{-1}$ soil in Barré et al.,
2010) and Grignon (7.12 *vs*. 6.80 gC.kg$^{-1}$ soil in Barré et al., 2010), but strongly modified (increased) at Ultuna (6.95 *vs*. 3.90 gC.kg$^{-1}$ soil in Barré et al., 2010) and Rothamsted (10.46 *vs*. 2.72 gC.kg$^{-1}$ soil in Barré et al., 2010, Table 1).

Our results obtained under four contrasted pedoclimates of Northwestern Europe indicate a minimum value of 5 gC.kg$^{-1}$ soil (lowest boundary of the four 95% confidence intervals, Table 1) and a maximum value of 11 gC.kg$^{-1}$ soil (highest boundary of the four 95% confidence intervals, Table 1) for $CP_{SOC}$ concentration in topsoils (0–20 to 0–25 cm depth). These estimates
are close, yet below the $CP_{SOC}$ concentration value of 12 gC.kg$^{-1}$ soil estimated by Buyanovsky and Wagner (1998b) for the topsoil (0–20 cm depth) of the Sanborn long-term (100 years) agronomic experiment (Columbia, Missouri, USA). Our estimates of topsoil $CP_{SOC}$ concentration are also well below the value of 16 gC.kg$^{-1}$ soil estimated by Franko and Merbach (2017) in the topsoil (0–30 cm depth) of the long-term (28 years) bare fallow experiment of Bad Lauchstädt (Central Germany). The soil type in Bad Lauchstädt (Haplic Chernozem) and its high concentration of slow-cycling black carbon
(estimated at 2.5 gC.kg$^{-1}$ soil, Brodowski et al., 2007) may explain this difference, as well as the relatively short time period under bare fallow (higher uncertainty on the inferred $CP_{SOC}$ concentration).

Among the wide range of $CP_{SOC}$ proportions (0.14 to 1) of our reference sample set, high values of $CP_{SOC}$ proportions (> 0.6) were obtained only for soils which had been under bare fallow for a long period of time: after several years or decades with negligible C inputs and sustained SOC decomposition (Table 1). Similarly, the low values of $CP_{SOC}$ proportions (< 0.25) of
our reference sample set were obtained for soils without vegetation but receiving high amounts of manure amendments at Versailles (Table 1). It could be argued that $CP_{SOC}$ proportion values obtained for bare soils with or without organic matter amendments may not be representative of $CP_{SOC}$ proportions of soils under conventional management practices. However, it



is interesting to notice that soils of the reference sample set with vegetation and experiencing classical management practices (grassland at Rothamsted, cropland at Ultuna) also showed a wide range of $CP_{SOC}$ proportions, from 0.25 to 0.56 (Table 1). Moreover, other studies have shown the high variability of $CP_{SOC}$ proportion in soils. For instance, Falloon et al. (1998) listed a series of published values of $CP_{SOC}$ proportions ranging from 0.13 to 0.59. More recently, Mills et al. (2014)

published a large dataset of $CP_{SOC}$ proportions in uncultivated topsoils (*ca.* 15 cm depth). They estimated $CP_{SOC}$ proportions using a global dataset of topsoil radiocarbon ($^{14}$C) data and a steady-state SOC turnover model with a fixed mean residence time of 1000 years for persistent SOC. Their estimates of $CP_{SOC}$ proportions varied greatly from 0.03 to 0.98 (mean = 0.48, standard deviation = 0.22, n = 232, soils with inconsistent negative modeled SOC pools values were removed), with significantly higher $CP_{SOC}$ proportions in non-forest than in forest uncultivated ecosystems (Mills et al., 2014).

Overall, those combined results illustrate the wide range of $CP_{SOC}$ concentrations and proportions in topsoils that may depend upon pedoclimate, land-use and management practices. Additionally, these results show the value of LTBF experiments to investigate the long-term dynamics of SOM.

## 4.2 A quantitative link between the long-term biogeochemical stability of SOC and its thermal stability and bulk chemistry

This work strengthens the existence of a link between SOC persistence in ecosystems and its thermal stability, providing evidence of the first quantitative link between thermal and *in-situ* long-term (> 100 years) biogeochemical SOC stability (Fig. 4). The regression model yields accurate RE6-RF predictions of $CP_{SOC}$ proportions with 95% prediction intervals that fully propagate the uncertainties originating from the calibration set that was used to build the model. Predictions on the validation set illustrate that the error propagation scheme provides highly conservative 95% prediction intervals of the $CP_{SOC}$

proportion in new samples, all intervals including their respective reference estimate of $CP_{SOC}$ proportion (Fig. 4). Despite rather large prediction intervals, the RE6-RF regression model clearly discriminates soils with small $CP_{SOC}$ proportions from samples with large $CP_{SOC}$ proportions (Fig. 4). This model based on RE6 thermal analysis can thus be used to predict the size of the $CP_{SOC}$ pool with known uncertainty in new soil samples from similar pedoclimates and with thermal characteristics (*i.e.* RE6 predictor variables) similar to those of the reference sample set.

Our results also illustrate the complex relationships between thermal analysis based parameters of SOC stability and the $CP_{SOC}$ proportion. The hypothesis behind the use of SOC thermal stability as a proxy of its biogeochemical stability implies positive correlations between the size of the $CP_{SOC}$ pool and temperature parameters derived from thermal analysis such as the 25 RE6 temperature parameters calculated in this study. Significant positive correlations with the $CP_{SOC}$ proportions were indeed found for the majority (15 out of 25) of RE6 temperature parameters, with very high and positive Spearman's ρ

values for some of them (Table 2). This was notably the case of the RE6 temperature parameter corresponding to 50% of $CO_2$ gas evolution during the oxidation stage, $T_{50\_CO2\_OX}$ that had been previously shown to systematically increase with bare fallow duration on the same soils by Barré et al. (2016). This study extends the results of Barré et al. (2016) towards a quantitative link between RE6 temperature parameters and SOC persistence (direct correlations and predictions of the size of



the CP$_{SOC}$ pool rather than time under bare fallow treatment). It also extends those results to non-bare fallow soils: bare soils receiving organic amendments (at Grignon and Versailles), cropland soils (Ultuna) and grassland soils (Rothamsted). Conversely, ten RE6 temperature parameters showed no significant correlation or significant negative correlations with the CP$_{SOC}$ proportion. Weak or negative correlations occurred principally for temperature parameters calculated on thermograms

of the pyrolysis stage of the RE6 analysis: for all parameters of the CO thermogram and low temperature parameters of the HC and CO$_2$ thermograms (Table 2). Negative correlations contradict the above-mentioned hypothesis, with the evolution of a similar proportion of the total amount of gases (HC pyrolysis effluents or CO) occurring at lower temperatures for samples with high CP$_{SOC}$ proportions than for soils with low CP$_{SOC}$ proportions. A possible explanation for this unexpected observation could be that the pyrolysis of SOC in samples with high proportion of CP$_{SOC}$ may undergo an enhanced

pyrolysis catalyst effect by soil minerals (Auber, 2009), which are relatively more abundant in those samples generally characterized by low total SOC concentrations.

Despite the fact that three RE6 parameters used here, *i.e.* the TLHC-index, the I-index, and the R-index, had originally been proposed as useful qualitative metrics of soil carbon dynamics, reflecting a proportion of thermally resistant or labile hydrocarbons (Disnar et al., 2003; Sebag et al., 2006; Saenger et al., 2013, 2015; Sebag et al., 2016), those parameters were

weakly correlated (TLHC-index) or not correlated (I-index, R-index) to the CP$_{SOC}$ proportion. Furthermore, they also had a weak importance in the random forests model predictions of the CP$_{SOC}$ proportion (Table 2). The poor link between those three RE6 parameters and the CP$_{SOC}$ proportion may be explained by the high residence time of CP$_{SOC}$ (> 100 years). Indeed, so far those parameters have been related to the proportion of SOC present in the particulate organic matter fraction (> 50 µm), a SOC pool characterized by a residence time in soils generally below 20 years (Saenger et al., 2015; Soucémarianadin

et al., 2018).

The two RE6 parameters reflecting SOC bulk chemistry showed highly significant correlations with the CP$_{SOC}$ proportion. This confirms, and extends to vegetated soils, the observed decreasing trend for HI and increasing trend for OI$_{RE6}$ (except at Versailles which has high proportions of pyrogenic carbon) with bare fallow duration observed by Barré et al. (2016) on the bare fallow treatments of the same experimental sites. Soils with high proportions of CP$_{SOC}$ are thus characterized by an

oxidized and H-depleted organic matter.

### 4.3 Perspectives to improve and foster RE6 thermal analysis based predictions of the size of the CP$_{SOC}$ pool

Future developments of this work must extend the Rock-Eval 6 thermal analysis regression model to a wider range of pedoclimates and to other biomes. As sites with LTBF treatments are not widespread, complementing the reference sample set may be achieved by using soils that have different soil forming factors (*e.g.* climate, parent material) and (i) which are

sampled from long-term (> 50 to 100 years) experiments with contrasted SOC inputs, enabling the estimation of their CP$_{SOC}$ concentration (Buyanovsky and Wagner, 1998a; 1998b), or (ii) for which the mean SOC age is known from radiocarbon data, enabling the estimation of the size of their persistent SOC pool (Trumbore, 2009; Mills et al., 2014).



Another development of this work will involve elucidating the fundamental mechanisms linking the biogeochemical stability of SOC with its thermal stability (see *e.g.* Leifeld and von Lützow, 2014). This was beyond the scope of this work, yet it remains an exciting field of research that should be addressed in the future, as highlighted by the unexpected observations discussed in Section 4.2 and by other recent works that found no relationships between the thermal oxidation of SOC

between 200 °C and 400 °C and the size of SOC pools with shorter residence times in soils (below or above *ca*. 18 years, Schiedung et al., 2017).

Overall, this work demonstrates the value of Rock-Eval 6 as a routine method for quantifying the size of the centennially persistent SOC pool with known uncertainty in temperate soils. The relatively low cost of the Rock-Eval 6 technique and the robustness of the thermal analysis regression model makes it possible to apply it to soil monitoring networks across a

continuum of scales, as a reliable proxy of SOC persistence. This may be part of the framework proposed by O'Rourke et al. (2015) to better understand SOC processes at the biosphere to biome scales, and should be added to the soil carbon cycling proxies recently listed by Bailey et al. (in press). Mapping persistent SOC at large scales may allow the identification of regional hotspots of centennially persistent SOC that may contribute little to climate change by 2100. It may also provide information on the sustainability of additional SOC storage from soil carbon sequestration strategies such as those promoted

by the international 4 per 1000 initiative in agriculture and forestry (https://www.4p1000.org/; Dignac et al., 2017; Minasny et al., 2017; Soussana et al., in press). A global map of persistent SOC based on this empirical RE6 thermal analysis model could also be useful for improving the parameterization of models simulating SOC dynamics (Falloon and Smith, 2000; Luo et al., 2014; He et al., 2016). The integration of large-scale information on the size of SOC kinetic pools may indeed provide an adequate complement to the global data sets on SOC fluxes that are currently under development and restructuration

(Hashimoto et al., 2015; Luo et al., 2016; Harden et al., in press).

**Acknowledgements**

This work was funded by ADEME and EC2CO (CARACAS project). Pierre Barré, Lauric Cécillon, Laure Soucémarianadin and Suzanne Lutfalla thank the financial support of the Mairie de Paris (Emergences Programme). We thank Thomas Eglin (ADEME) for his valuable suggestions on this work. We thank INRA and AgroParisTech for access to and maintenance of

the Versailles 42 plots and the Grignon 36 plots long-term experiments and their sample archives. We thank Rothamsted Research and the Lawes Agricultural Trust for access to archived samples and the BBSRC for support under the Institute National Capabilities program grant (BBS/E/C/000J0300). We gratefully acknowledge the Faculty of Natural Resources and Agricultural Sciences for providing funds for maintenance of the Ultuna long-term field experiment and for its sample archive.





**Supplementary material**

(see Supplementary material manuscript)

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





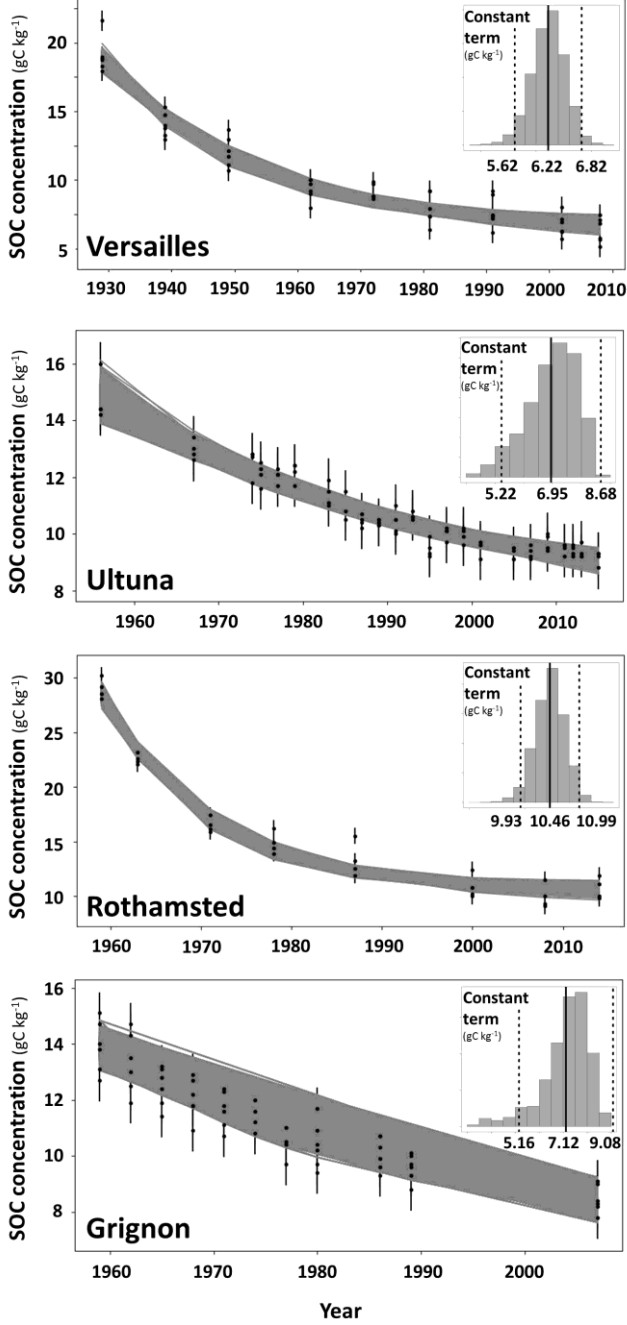

**Figure 2: Evolution of SOC concentration (gC.kg$^{-1}$ soil) with time for the bare fallow plots of each experimental site, and representation of the $3 \times 10^4$ fitted exponential decay functions (Bayesian curve fitting method) from which a site-specific CP$_{SOC}$ concentration (model parameter c) and its 95% confidence interval were determined (histogram in the upper right side of each scatter plot). At each site, the 95% confidence interval around the CP$_{SOC}$ concentration was determined as c ± 1.96 sd(c), where c is the model parameter c in Eq. (1) and sd(c) is its standard deviation calculated in Eq. (6).**





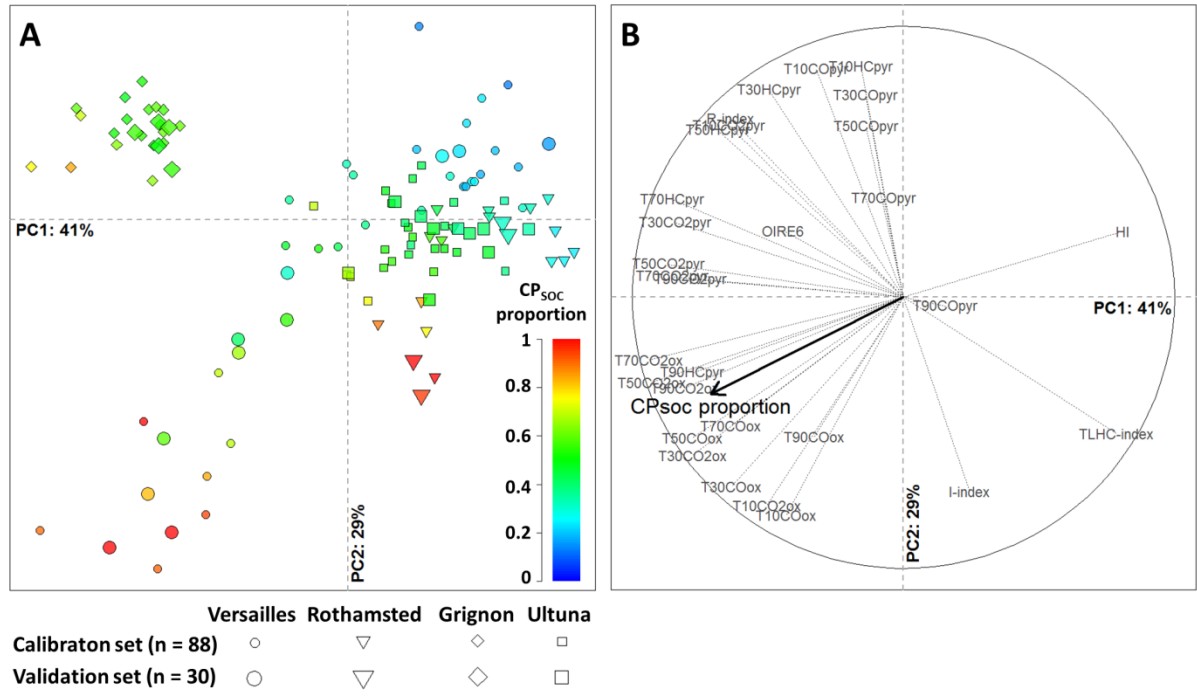

**Figure 3: Principal component analysis (PCA) of the 30 RE6 parameters of the calibration soil sample set (n = 88). A: the scores of the calibration samples on the first two principal components are represented in the factorial map, as well as the projected principal component scores of the validation samples (n = 30). A color scale is used to represent the CP$_{SOC}$ proportion (determined using Eq. (2)) in all samples. B: PCA loadings of the 30 RE6 parameters and projection of the CP$_{SOC}$ proportion variable in the PCA correlation circle.**







**Figure 4: Performance of the random forests regression model based on Rock-Eval 6 thermal analysis (RE6-RF) for predicting the CP$_{SOC}$ proportion. The performance statistics on the calibration set (n = 88) and on the validation set (n = 30) of the RE6-RF multivariate regression model are shown. Horizontal bars represent the estimated uncertainty (95% confidence intervals) on the reference CP$_{SOC}$ proportion values of the validation set, calculated as: CP$_{SOC}$ proportion [sample] ± 1.96 × sd(CP$_{SOC}$ proportion [sample]). Vertical bars represent the propagated errors (95% confidence intervals) on the RE6-RF predicted CP$_{SOC}$ proportion values of the validation sample set, calculated as $\bar{\bar{y}} \pm \hat{\tau} \times$ sd($\hat{y}$) (see Section 2.3.5), with a $\hat{\tau}$ value of 2.10 (Fig. S1). Abbreviations: RMSEC, root-mean-square error of calibration; RMSEP, root-mean-square error of prediction; RPD, ratio of performance to deviation; sd, standard deviation.**





**Table 1: Measured total SOC concentrations, estimated site-specific $CP_{SOC}$ concentrations, and resulting $CP_{SOC}$ proportions in four long-term agronomic experimental sites used to generate calibration and validation soil sample sets. Abbreviations: LTBF, long-term bare fallow; min, minimum; max, maximum; sd, standard deviation; CI, confidence interval.**

| Site | Treatments (number of samples) | SOC concentration ($gC.kg^{-1}$ soil) mean (min, max, sd) | $CP_{SOC}$ concentration ($gC.kg^{-1}$ soil) mean (95 % CI) | $CP_{SOC}$ proportion mean (min, max, sd) |
|---|---|---|---|---|
| **Versailles** | Manure (n = 20) | 27.9 (17.1, 45.5, 8.2) | 6.22 (5.62–6.82) | 0.24 (0.14, 0.36, 0.07) |
| | LTBF (n = 20) | 10.5 (5.4, 19.7, 4.4) | | 0.67 (0.32, 1.00, 0.24) |
| **Rothamsted** | Grassland (n = 8) | 36.8 (31.8, 42.6, 4.8) | 10.46 (9.93–10.99) | 0.29 (0.25, 0.33, 0.04) |
| | LTBF (n = 12) | 17.7 (9.7, 30.5, 7.5) | | 0.68 (0.34, 1.00, 0.25) |
| **Ultuna** | Cropland (n = 23) | 15.8 (12.4, 20.3, 2.2) | 6.95 (5.22–8.68) | 0.45 (0.34, 0.56, 0.06) |
| | LTBF (n = 11) | 12.0 (9.1, 16.3, 2.4) | | 0.60 (0.43, 0.76, 0.12) |
| **Grignon** | Straw or composted straw (n = 12) | 12.9 (11.7, 14.2, 0.8) | 7.12 (5.16–9.08) | 0.55 (0.50, 0.60, 0.03) |
| | LTBF (n = 12) | 11.8 (8.4, 14.7, 1.9) | | 0.62 (0.48, 0.85, 0.11) |
| **Calibration set** (n = 88) | | 18.0 (5.5, 45.5, 9.5) | | 0.50 (0.14, 1.00, 0.21) |
| **Validation set** (n = 30) | | 16.1 (5.4, 38.8, 7.9) | | 0.53 (0.16, 1.00, 0.23) |
| **All samples** (n = 118) | | 17.5 (5.4, 45.5, 9.1) | | 0.51 (0.14, 1.00, 0.21) |



**Table 2: Spearman's rank correlation coefficient test between the 30 RE6 parameters and the CP$_{SOC}$ proportion, and variable importance (ranking) of the 30 RE6 parameters to predict CP$_{SOC}$ proportion in the random forests model based on Rock-Eval 6 thermal analysis (RE6-RF, calibration soil sample set, n = 88). Symbols for *p*-values: \*\*\* *p* < 0.001; \*\* *p* < 0.01; \* *p* < 0.05; NS *p* > 0.05 = not significant.**

| RE6 parameter | Spearman's $\rho$ with CP$_{SOC}$ proportion | *p*-value | Variable importance to predict CP$_{SOC}$ proportion in the RE6-RF regression model (rank) |
|---|---|---|---|
| T$_{10\_HC\_PYR}$ | -0.42 | \*\*\* | 15 |
| T$_{30\_HC\_PYR}$ | -0.11 | NS | 29 |
| T$_{50\_HC\_PYR}$ | 0.11 | NS | 26 |
| T$_{70\_HC\_PYR}$ | 0.38 | \*\*\* | 21 |
| T$_{90\_HC\_PYR}$ | 0.72 | \*\*\* | 8 |
| T$_{10\_CO\_PYR}$ | -0.22 | \* | 6 |
| T$_{30\_CO\_PYR}$ | -0.39 | \*\*\* | 12 |
| T$_{50\_CO\_PYR}$ | -0.34 | \*\* | 16 |
| T$_{70\_CO\_PYR}$ | -0.23 | \* | 20 |
| T$_{90\_CO\_PYR}$ | -0.08 | NS | 22 |
| T$_{10\_CO2\_PYR}$ | 0.03 | NS | 9 |
| T$_{30\_CO2\_PYR}$ | 0.72 | \*\*\* | 5 |
| T$_{50\_CO2\_PYR}$ | 0.81 | \*\*\* | 1 |
| T$_{70\_CO2\_PYR}$ | 0.77 | \*\*\* | 3 |
| T$_{90\_CO2\_PYR}$ | 0.66 | \*\*\* | 23 |
| T$_{10\_CO\_OX}$ | 0.51 | \*\*\* | 4 |
| T$_{30\_CO\_OX}$ | 0.71 | \*\*\* | 17 |
| T$_{50\_CO\_OX}$ | 0.64 | \*\*\* | 27 |
| T$_{70\_CO\_OX}$ | 0.42 | \*\*\* | 28 |
| T$_{90\_CO\_OX}$ | 0.14 | NS | 30 |
| T$_{10\_CO2\_OX}$ | 0.72 | \*\*\* | 11 |
| T$_{30\_CO2\_OX}$ | 0.83 | \*\*\* | 2 |
| T$_{50\_CO2\_OX}$ | 0.82 | \*\*\* | 7 |
| T$_{70\_CO2\_OX}$ | 0.80 | \*\*\* | 10 |
| T$_{90\_CO2\_OX}$ | 0.54 | \*\*\* | 18 |
| I-index | 0.13 | NS | 24 |
| R-index | 0.05 | NS | 25 |
| TLHC-index | -0.24 | \* | 19 |
| HI | -0.78 | \*\*\* | 14 |
| OI$_{RE6}$ | 0.42 | \*\*\* | 13 |