# Peer review of "A model based on Rock-Eval thermal analysis to quantify the size of the centennially persistent organic carbon pool in temperate soils"

_Biogeosciences, 2018_

## Referee Comment (RC1) · Anonymous Referee #1 · 26 Feb 2018

This manuscript presents an analysis of thermal properties of soil organic matter (SOM) and how they relate to the concept of persistence. The analysis is performed on a range of long-term bare follow experiments, which are of special significance to understand long-term carbon dynamics in soils. The manuscript is well written and most of the information is well presented. I only have two major conceptual issues, but overall no technical comments.

[Figure]

**1 Major comments**

One important source of confusion in this manuscript is the use of the concept *residence time*. Notice that in soils one must distinguish between the concepts of age and transit(residence) time (Bruun et al., 2004; Manzoni et al., 2009; Derrien and Amelung, 2011). What the authors are trying to estimate here is an indication of the *age* of the SOM, not the residence time. These should be more clearly treated in the introduction and the discussion. Currently, the use of these terms is ambiguous.

The other important issue in this manuscript is also conceptual. The model presented in equ. 1 and used to compute the centennially persistent pool is, in my opinion, inappropriate. It assumes that an amount of 'inert' carbon $c$ is sitting there doing nothing and it will never decompose. This is highly unlikely, because there's always some small probability that carbon that is stabilized either by mineral association or protection in aggregates, would get consumed by microorganisms and respired as $CO_2$.

One possibility to deal with this issue is to modify the model of equation 1 to account for this small probability of decay of the centennial pool. This can be achieved by simply adding a second decay term so,

$$\gamma(t) = ae^{-b_1t} + ce^{-b_2t} \tag{1}$$

However, this equation adds an additional parameter to estimate. To solve this problem, notice that at time $t = 0$, $\gamma(t = 0) = a + c = \gamma_0$. So, we can modify this equation as

$$\gamma(t) = \gamma_0\alpha e^{-b_1t} + \gamma_0(1 - \alpha)e^{-b_2t} \tag{2}$$

Now we have an equation with the same number of parameters to estimate as the original one, but with the ability to conceptually add a probability of the carbon in the persistent pool to be decomposed over time. The size of this persistent pool would

be simply $(1 - \alpha)$, expressed as a proportion of the initial amount of carbon at the beginning of the experiment.

**References**

Bruun, S., Six, J., and Jensen, L. S. (2004). Estimating vital statistics and age distributions of measurable soil organic carbon fractions based on their pathway of formation and radiocarbon content. *Journal of Theoretical Biology*, 230(2):241–250.

Derrien, D. and Amelung, W. (2011). Computing the mean residence time of soil carbon fractions using stable isotopes: impacts of the model framework. *European Journal of Soil Science*, 62(2):237–252.

Manzoni, S., Katul, G. G., and Porporato, A. (2009). Analysis of soil carbon transit times and age distributions using network theories. *J. Geophys. Res.*, 114.

---

## Referee Comment (RC2) · Anonymous Referee #2 · 27 Feb 2018

The manuscript presents an important and innovative contribution to the quantitative determination of the SOC pool with centennial turnover times (CPSOC). Quantitative understanding of this pool is highly important, e.g. for model initialization. The paper is well written and the statistical methods used are state of the art and without major flaws, as far as I can judge. However, there is one major point, in which the authors have not yet entirely convinced me that the derived model is capable to predict CPSOC in "new" samples:

1. Machine learning is used to find the best regression model predicting the proportion of CPSOC. A high R2 (0.91) in the calibration dataset is impressive and shows that

the thermal stability (RE6) can be linked to biogeochemical stability, which has been shown before. Now, the interesting thing is the validation: the authors report the same R2 for the validation set and show that they scatter as nicely along the 1:1 line as the calibration dataset does: Of course, this is the case because the dataset was randomly split, although the samples were not independent but originated from only 4 experiments. So the question is, will the results we similarly good, if for example 3 sites are used to train the model and 1 site is used for validation? This would give a much more honest picture on the validity of the approach. I guess that the prediction would not be as good: According to the Barre et al 2016 paper, at least the three presented thermal stability parameters HI, OI and T50CO2_ox which played an intermediate to important role also in the present study, varied considerably across sites. Also Figure 3A indicates that the 'thermal signature' of the samples is really site dependent. So for me the question is: Can this product really hold what the authors are promising, e.g. in the last sentence of the abstract: 'This model can thus be used to predict. . .'? This has to be clarified and if not the case be discussed with much more caveats. Uncertainties are already huge and they would probably inflate if new samples shall be predicted.

I also have some (partly minor) specific comments:

P5,line 8: Why such a huge intercept in the regression (0.4)?

P6, line 23: Fixed standard deviation for SOC concentration data? Why, and what is it exactly derived from?

P12, line 28: I thought it was 30 RE6 parameters, here it says it was 25?

Discussion:

1. Is very technical and focused on the specific RE6 method and related papers. Authors miss the chance to broaden the perspective and discuss this approach to estimate CPSOC in comparison to other approaches or to establish a clear link to kinetic models.

2. Is very positive about the overall results (and sample set), and although uncertainty was a clearly stated focus of the study it is not really taken up here: Yes, the sample set is truly unique, but this is also the problem: How uncertain will the CPSOC estimation in soils be, which do not have bare fallow treatments?

———————————————————

---

## Short Comment (SC1) · 20 Mar 2018

In their paper, the authors use thermograms resulting from thermal analyses, i.e. the pyrolysis and oxidation phases of Rock-Eval pyrolysis, to quantify the size of the centennially persistent organic carbon pool in temperate soils. They calculate a large range of variables that are further used to forecast the size of this SOC pool. These variables fall into two categories.

The first category gathers temperatures corresponding to the integration of the cumulative frequency curve up to a given (arbitrary) threshold value. It encompasses 25 variables corresponding to a series of five threshold values (i.e. 10, 30, 50, 70 and

90%) applied to the five thermograms obtained during measurements (i.e. S2, S3, S3', S4, S5). This approach is no more than a simple and fractional discretization of thermogram shapes.

The second category includes indices corresponding to the integration of the gas flow curve. It contains five variables including two standard parameters related to the composition of SOM (HI, OI) and three indices calculated solely from S2 pyrograms (flows of HC measured during the pyrolysis phase). These last three parameters have been constructed in order to characterize the whole shape (and information) of thermograms by taking into account the most labile thermal fraction (i.e. T <400-450 C). Unfortunately, the authors misuse them as they pretend to apply these indices in order to characterize the size of the centennially persistent organic carbon pool in samples from bare fallows. Moreover, their results show that these indices are not relevant (because obviously wrongly applied); their explanation stresses the sensitivity to the OM's most labile fraction of these indices, which is exactly their goal.

Consequently, it may have been preferable to limit their comparisons to the previous 25 variables proposed, and to explain why the other indices cannot be considered as relevant. However, if their idea is the introduction of some elements of comparison, it would undoubtedly be more interesting to propose other indices, such as the relative surface (or the part of HI corresponding to the relative surface) of the flow curve above certain threshold values (e.g. 460 C and 520 C), which correspond to the most stable thermal fraction of carbon pools.

---

## Author Comment (AC1) · 23 Mar 2018

We thank Reviewer 1 for his/her stimulating and constructive comments on our manuscript.

Major point 1: use of the concept of residence time in the manuscript

(1) Comment from Referee 1

One important source of confusion in this manuscript is the use of the concept residence time. Notice that in soils one must distinguish between the concepts of age and transit(residence) time (Bruun et al., 2004; Manzoni et al., 2009; Derrien and Amelung,

2011). What the authors are trying to estimate here is an indication of the age of the SOM, not the residence time. These should be more clearly treated in the introduction and the discussion. Currently, the use of these terms is ambiguous.

(2) Response to Referee 1's comment

We agree with Reviewer 1 that it is important to distinguish between the concepts of age and residence time of organic carbon in soils. However, our methodology does not focus on either the estimation of the age nor on the estimation of the residence time of soil organic carbon (SOC). Our methodology aims at estimating the size of the centennially persistent SOC (CPsoc) pool, a SOC pool whose mean age and mean residence time are both assumed to be high (e.g. several centuries) but for which precise definitions are not necessary. The only required property for defining the CPsoc pool is not its mean age or its mean residence time, but the non-significant change in its size in periods inferior to the century. Specifically, we defined the CPsoc pool (gC.kg-1 soil) at each site as the constant term of an exponential plus constant model fitted on the temporal evolution of SOC under bare fallow treatment. The only assumption that we made was that, given our data set (temporal evolution of SOC under 5 to 8 decades of bare fallow treatment or associated non-bare fallow treatment), the size of the CPsoc pool remained constant (i.e. did not change significantly, see also our response to major point 2 below).

(3) Proposed changes in manuscript

Overall, we thus did not make any specific estimation of the age or the residence time of the CPsoc pool, but we propose to treat this point more clearly in a revised version of our manuscript (introduction and discussion sections) to avoid any confusion regarding the concepts of age and residence time of organic carbon in soils.

Major point 2: centennially persistent soil organic carbon is not inert

(1) Comment from Referee 1

The other important issue in this manuscript is also conceptual. The model presented in equ. 1 and used to compute the centennially persistent pool is, in my opinion, inappropriate. It assumes that an amount of 'inert' carbon c is sitting there doing nothing and it will never decompose. This is highly unlikely, because there's always some small probability that carbon that is stabilized either by mineral association or protection in aggregates, would get consumed by microorganisms and respired as $CO_2$.

(2) Response to Referee 1's comment

We fully agree with Reviewer 1 that organic carbon is not biogeochemically inert in soils vis-à-vis microbial decomposition, and that the CPsoc pool is also (though very slowly) gaining and loosing carbon. However, regarding the CPsoc pool, its high residence time (e.g. several centuries; not precisely defined here, see above) masks changes in its size at the time scale of our sample set (5 to 8 decades). We thus pragmatically and operationally considered the CPsoc pool to be mathematically inert (the constant term in an exponential plus constant model) given our data set, though we agree that the CPsoc pool is a biogeochemically stable but not inert SOC pool. We argue that since all models are simplifications of reality, assuming an inert SOC pool is acceptable for a SOC pool with very low decomposition rate, as implemented in many widely used SOC models such as RothC. We have considered performing a double exponential (without constant) as suggested by Reviewer 1, but we think that a constant is mathematically the most appropriate way to model the CPsoc pool with our data (maximum 80 years of decomposition). Furthermore, we think that the argument that a double exponential model does not add parameters is not true: $y(t)=y_0*a*exp(-k_1*t) + y_0*(1-a)*exp(-k_2*t)$ has 3 parameters – but $y(t)= a + y_0*(1-a)*exp(-k_1*t)$ has only 2. Parsimony (Ockham's razor) and equifinality of more complex model lead us to propose keeping a single exponential plus constant model to estimate the size of the CPsoc pool (gC.kg-1 soil) in the bare fallow treatments of each study site.

(3) Proposed changes in manuscript

We propose to revise the manuscript to clarify that the CPsoc pool is not biogeochemically inert in our view, but that it is mathematically more sound to mathematically simulate it as a constant on such a dataset.

---

## Author Comment (AC2) · 23 Mar 2018

We thank Reviewer 2 for his/her constructive and useful comments on our manuscript.

Major point 1: Validation of the multivariate regression model to predict the size of the centennially persistent SOC (CPsoc) pool in "new" soils

(1) Comment from Referee 2

Machine learning is used to find the best regression model predicting the proportion of CPsoc. A high $R^2$ (0.91) in the calibration dataset is impressive and shows that the thermal stability (RE6) can be linked to biogeochemical stability, which has been

shown before. Now, the interesting thing is the validation: the authors report the same $R^2$ for the validation set and show that they scatter as nicely along the 1:1 line as the calibration dataset does: Of course, this is the case because the dataset was randomly split, although the samples were not independent but originated from only 4 experiments. So the question is, will the results we similarly good, if for example 3 sites are used to train the model and 1 site is used for validation? This would give a much more honest picture on the validity of the approach. I guess that the prediction would not be as good: According to the Barre et al 2016 paper, at least the three presented thermal stability parameters HI, OI and T50CO2_ox which played an intermediate to important role also in the present study, varied considerably across sites. Also Figure 3A indicates that the 'thermal signature' of the samples is really site dependent. So for me the question is: Can this product really hold what the authors are promising, e.g. in the last sentence of the abstract: 'This model can thus be used to predict...'? This has to be clarified and if not the case be discussed with much more caveats. Uncertainties are already huge and they would probably inflate if new samples shall be predicted.

(2) Response to Referee 2's comment

We randomly split our sample set into a calibration and a validation set, therefore including soil samples from the four study sites in both the calibration and the validation set. We thus agree with Reviewer 2 that the validation of the multivariate regression model is not based on truly independent soil samples, even though samples from the validation set were not used in the calibration set. As discussed in the manuscript, the good fit obtained for the current validation set (Figure 4 in the manuscript) indicates that the multivariate regression model can be used to predict the size of the CPsoc pool with a known uncertainty in soils with pedoclimates similar to those found in the four study sites (Versailles, Grignon, Rothamsted and Ultuna; Supplementary material S1).

However, Reviewer 2 asks an important question: "will the results be similarly good, if for example 3 sites are used to train the model and 1 site is used for validation?"

Since each site used in this study has a specific pedoclimate (even the two sites with a similar climate, Versailles and Grignon, have different soil mineralogy, with carbonate soils in Grignon and soils developed in loess in Versailles), we have to slightly rephrase Reviewer 2's question regarding the specificity of our data set. In fact, Reviewer 2 asks if the multivariate regression model is able to predict the size of the CPsoc pool in soils with a different pedoclimate (i.e. a pedoclimate not included in the calibration set). Or more generally, can the model predict the CPsoc proportion outside of the studied sites?

We agree with Reviewer 2 that testing the multivariate regression model sensitivity to pedoclimate (i.e. by validating it on a site with a new pedoclimate) would provide useful information to the readers regarding its applicability on soils from different pedoclimates.

We argue that a necessary prerequisite for applying the multivariate regression model on "new" soil samples from a different pedoclimate is the thermal similarity between the "new" soil samples and samples of the calibration set (i.e. similar range of values for the 30 Rock-Eval parameters that were used as predictors in the multivariate regression model).

All soils from Grignon (carbonate site), and some samples from Versailles and Rothamsted show some important specificities regarding their thermal characteristics, while all soils from Ultuna had thermal characteristics similar to some samples from Versailles and Rothamsted (see the PCA plot in Figure 3 in the manuscript).

(3) Proposed changes in manuscript

We therefore propose to add a second validation scheme to test the sensitivity of the multivariate regression model to a different pedoclimate. Ultuna can be used as a truly independent validation site with similar thermal characteristics but a different pedoclimate than the calibration set.

The results of the predictions for Ultuna samples, using samples from Versailles, Grignon and Rothamsted for calibration of the multivariate regression model are shown in Figure 1 (see below). As expected, the $R^2$ strongly decreases, yet the error of prediction of the model does not increase strongly (RMSEP = 0.09 vs. 0.07 in the previous validation scheme).

Overall, these new results illustrate the sensitivity of the multivariate regression model to a very different pedoclimate (different climate and soil mineralogy), yet they clearly show the potential of the model based on Rock-Eval analysis for predicting the proportion of CPsoc in "new" soil samples. We thus propose to include and discuss them thoroughly in a revised version of the manuscript.

Specific comments:

(1) Comment from Referee 2

P5, line 8: Why such a huge intercept in the regression (0.4)?

(2) Response to Referee 2's comment

The relatively high value of the intercept may be linked to the fact that when estimating the total organic carbon content, a small amount of organic carbon is not taken into account by the commercial software of Rock-Eval 6 (organic carbon being volatilized as CO or CO2 at high temperatures, that may be inorganic carbon in carbonated soils). Underestimation of SOC concentration by RE6 has already been reported (e.g. Saenger et al., 2013). As soils from Grignon contain carbonates, we chose the same metric of SOC_RE6 for all samples, even if they are slightly biased towards lower values.

(1) Comment from Referee 2

P6, line 23: Fixed standard deviation for SOC concentration data? Why, and what is it exactly derived from?

(2) Response to Referee 2's comment

We used a fixed value for the standard deviation for SOC concentration data obtained from elemental analyzer (SOC_EA; 0.75 gC.kg-1 soil). As stated in the manuscript, this value is a conservative estimate of the standard deviation of SOC_EA data estimated by Barré et al. (2010) for the same soils. In the latter paper, the authors stated that "standard deviation [was] estimated from 15 replicate determinations of C in soil samples taken from the same plot in Grignon in 1959 (Barré et al., 2010). The measured standard deviation was 0.3 gC.kg-1. As the C contents at the different LTBF sites were determined on composite samples from the same plot at each site, it was considered that the a priori error on measurements should be less than 0.5 gC.kg-1 for every site". They finally applied a standard deviation of 0.5 gC.kg-1 for every site except Versailles, where the final standard deviation for SOC concentration data was 0.75 gC.kg-1 (Barré et al., 2010).

(1) Comment from Referee 2

P12, line 28: I thought it was 30 RE6 parameters, here it says it was 25?

(2) Response to Referee 2's comment

We have indeed used a total of 30 RE6 parameters in the multivariate regression model, but only 25 of them are RE6 temperature parameters (i.e. unit: °C). In this section of the manuscript (P12, line 28), we only discuss the outcomes of the 25 temperature parameters. We discuss the outcomes of the remaining 5 RE6 parameters (TLHC-index, I-index, R-index, HI and OIRE6) later in the manuscript (P13, lines 12-25).

(1) Comment from Referee 2

Discussion 1: Is very technical and focused on the specific RE6 method and related papers. Authors miss the chance to broaden the perspective and discuss this approach to estimate CPsoc in comparison to other approaches or to establish a clear link to

kinetic models.

(2) Response to Referee 2's comment

Only one out of three sections of the discussion (section 4.2) is devoted to the discussion of the specific RE6 method and related papers. We agree that this section 4.2 is rather technical but we think that critically discussing some technical limitations of our approach is necessary. Section 4.1 of the discussion already broadens the scope of our study and discusses different methodologies that have been used to produce estimates of the CPsoc concentration under various pedoclimatic conditions (i.e. in other long term agronomic experiments or using other analytical techniques and/or models such as radiocarbon (14C) data and steady-state SOC turnover model).

We agree with Reviewer 2 that discussing our approach to estimate CPsoc in comparison to other approaches may be useful to the reader.

(3) Proposed changes in manuscript

We propose to add some references to alternative techniques used for initializing the size of SOC kinetic pools in models of SOC dynamics (e.g. Falloon et al., 1998; Zimmermann et al., 2007) in section 4.3 of the discussion.

(1) Comment from Referee 2

Discussion 2: Is very positive about the overall results (and sample set), and although uncertainty was a clearly stated focus of the study it is not really taken up here: Yes, the sample set is truly unique, but this is also the problem: How uncertain will the CPsoc estimation in soils be, which do not have bare fallow treatments?

(2) Response to Referee 2's comment and (3) Proposed changes in manuscript

Following our response to the major point raised by Reviewer 2 (see above), we propose to revise the manuscript to discuss in section 4.3 the new results regarding the multivariate regression model sensitivity to pedoclimate (i.e. prediction on "new" soils

from a different pedoclimate).

References:

Barré, P., Eglin, T., Christensen, B.T., Ciais, P., Houot, S., Kätterer, T., van Oort, F., Peylin, P., Poulton, P.R., Romanenkov, V., and Chenu, C.: Quantifying and isolating stable soil organic carbon using long-term bare fallow experiments. Biogeosciences, 7, 3839–3850, 2010

Falloon, P., Smith, P., Coleman, K., and Marshall, S.: Estimating the size of the inert organic matter pool from total soil organic carbon content for use in the Rothamsted carbon model. Soil Biol. Biochem., 30, 1207–1211, 1998.

Saenger, A., Cécillon, L., Sebag, D., and Brun, J.J.: Soil organic carbon quantity, chemistry and thermal stability in a mountainous landscape: A Rock-Eval pyrolysis survey. Org. Geochem., 54, 101–114, 2013.

Zimmermann, M., Leifeld, J., Schmidt, M.W.I., Smith, P., Fuhrer, J.: Measured soil organic matter fractions can be related to pools in the RothC model. Eur. J. Soil Sci., 58, 658–667, 2007.
* * *
[Figure]

[Figure]

**Fig. 1.** Performance of the random forests regression model based on Rock-Eval 6 thermal analysis (RE6-RF) for predicting the CPsoc proportion in soil samples from a new pedoclimate

---

## Author Comment (AC3) · 23 Mar 2018

We thank David Sebag for his Short Comment on our manuscript.

However, we disagree with his comments.

We think that all Rock-Eval 6 (RE6) parameters (both temperature parameters and indices corresponding to the integration of the gas flow curve) are fully eligible as potential predictors of the centennially persistent soil organic carbon (CPsoc) proportion in a non-parametric multivariate regression model. Indeed, even RE6 parameters associated to a thermally labile soil organic carbon (SOC) fraction, such as the I-index

and the TLHC-index, could have significant negative correlations with the CPsoc proportion. They would thus have been important predictors in the random forests model.

We thus argue that we did not "misuse" or "wrongly applied" any of the 30 RE6 parameters in the multivariate regression model.

Furthermore, we do not agree with David Sebag when he states that similarly to the I-index and the TLHC-index, the R-index takes "into account the most labile thermal fraction (i.e. T < 400-450 °C)". The R-index has been designed to represent the thermally stable SOC fraction (i.e. T > 400 °C): the "R-index [is] highlighting the contribution of the most refractory fraction or persistent SOM" (Sebag et al., 2016). Therefore, we think that it was particularly appropriate to use the R-index as a predictor of the CPsoc proportion in the random forests model.

Reference:

Sebag, D., Verrecchia, E.P., Cécillon, L., Adatte, T., Albrecht, R., Aubert, M., Bureau, F., Cailleau, G., Copard, Y., Decaens, T., Disnar, J.R., Hetényi, M., Nyilas, T., and Trombino, L.: Dynamics of soil organic matter based on new Rock-Eval indices. Geoderma, 284, 185–203, 2016.
* * *

---

## Author Response (AR2)

Dear Editor,

Thanks for evaluating positively our revised manuscript and for your advices for improving it.

We have now modified the revised manuscript according to your suggestions. We also carefully checked the manuscript again to make sure the English is correct.

We describe and explain below the different changes in the modified version of the revised manuscript. Our responses are indicated in blue while the locations of changes in the modified version of the revised manuscript are indicated in red.

We also specified (in the text and in captions of Figures and Tables) when the calibration set referred to a set with soils from all sites, to avoid confusion with the fully independent calibration set containing soils from the site of Ultuna only. (p.7 l.16, p.7 l.25–26, p.10 l.14, p.10 l.25, p.11 l.9 l.10 l.27, p.23 l.3 l.5, p.24 l.3 l.4, p.26 l.3 l.4, p.27 l.4)

We are looking to hearing from you.

Yours sincerely,

Lauric Cécillon

**Technical corrections**

Page 2, line 18: carbon input without the s.

Done (p.2 l.18)

Page 4, lines 25-31: I think centuries without the several might be long enough. The sentence starting with as a results seems a long, perhaps you ca rewrite it more concise? (As a result, its decline is minimal at the timescale of this study and we thus considered the centennially persistent SOC pool at each experimental site to be constant. Or something similar.)

Done (p.4 l.27–29)

Page 10, line 13: ….showed very different even contrasting correlations…

Done (p.10 l.13)

Page 11, Line 1 and 2: do you mean specific as in unique? Different from the others? If so maybe rewrite it a little, otherwise state what is specific about them.

Done (p.11 l.3–4)

Page 11, line 11: from the calibration set

Done (p.11 l.13)

[revised manuscript text omitted]